# OpenReview forum: "Layered Contextual Alignment: Multi-Agent Coordination for Web Automation Through Hierarchical Preference Learning"
_ICLR.cc/2026/Conference — ICLR 2026 Conference Withdrawn Submission_

### Official Review · Reviewer_ZP5L · 2025-10-23

**Soundness:** 1
**Presentation:** 1
**Contribution:** 1
**Rating:** 2
**Confidence:** 4

**Summary:**

The paper “Layered Contextual Alignment: Multi-Agent Coordination for Web Automation through Hierarchical Preference Learning” proposes a hierarchical coordination framework for multi-agent web automation. The approach introduces three contextual layers — global, shared, and individual — designed to enable emergent coordination without explicit communication

The paper fails to meet ICLR standards of theoretical rigor, empirical reliability, and presentation quality. Considering the accumulation of typos, poor figure design, inconsistent claims, and insufficient experimental justification,

I do not recommend acceptance for this ICLR review cycle.

**Strengths:**

The paper introduces the idea of hierarchical preference alignment for multi-agent coordination, attempting to balance autonomy and coherence through layered context embeddings.

Claims of production-level testing and throughput (10,000+ pages daily) demonstrate intent toward applied utility, even if details are missing.

**Weaknesses:**

The paper contains multiple typos (e.g., “tribution” in Figure 2), suggesting insufficient proofreading.

Figures are unclear and poorly formatted — inconsistent fonts, low resolution, and confusing layouts hinder understanding of the pipeline.

Experimental design lacks soundness — benchmarks are self-constructed, statistical reporting is questionable, and results are not validated on standard tasks like WebArena or GAIA.

**Questions:**

Q1: In Figure 1, is “Tae) Infra block” a typographical error? If not, please clarify its meaning. Additionally, what does the horizontal line on the left side of the Agent Message Protocol module represent?

Q2: Why does Figure 1 use a raster (non-vector) image instead of a high-resolution PDF or vector graphic? The current rendering is visually unclear and fails to effectively communicate the overall pipeline structure.

Q3: In Figure 2, is “task tribution” a typo for “task distribution”? Also, why are the fonts inconsistent across elements? The figure appears to be AI-generated, which does not meet ICLR’s expected presentation standards.

Q4: The abstract claims the system operates “without explicit communication,” yet the shared context layer clearly functions as a communication medium between agents. Could you reconcile this apparent conceptual contradiction?

Q5: The paper omits a citation for LangGraph and lacks engagement with recent research in multi-agent reinforcement learning — particularly in areas such as MARL stability and emergent communication. Could the authors expand on these related works?

Q6: Could you clarify which metric was used to compute the reported Cohen’s d = 6.39?
If the effect size was calculated based solely on Time (s), why were other key performance metrics (e.g., Success Rate and Quality) excluded from the effect size analysis?
Moreover, a Cohen’s d of 6.39 is extraordinarily large and highly unusual — please elaborate on how this value was obtained.

Q7: Why did you construct your own benchmark for evaluation instead of using established public environments such as WebArena or GAIA? Using standard benchmarks would enhance comparability and credibility.

Q8: The paper claims production-level deployment but provides no quantitative report of deployment costs, computational load, or token usage. Could you include these details for completeness?

Q9: The paper claims convergence guarantees but does not specify whether the underlying optimization problem is convex. Could you clarify whether your system’s objective function admits a well-defined global optimum?

Q10: In Appendix B, you reference mean-field theory and report a critical exponent of $β ≈ 0.5$.
Please justify rigorously how your multi-agent coordination dynamics mathematically map to the Landau–Ginzburg or any standard mean-field framework.
Does this analogy rely on any formal isomorphism, or is it merely a qualitative resemblance based on the shape of the performance curves?

---

### Official Review · Reviewer_DsLQ · 2025-10-30

**Soundness:** 1
**Presentation:** 1
**Contribution:** 1
**Rating:** 0
**Confidence:** 4

**Summary:**

The paper presents Layered Contextual Alignment (LCA) for multi-agent web automation, claiming coordination "without explicit communication". This central claim is directly contradicted by the paper's own architecture, which includes a "central coordinator" and "message protocol". The experimental claims, such as 100% success against "CAPTCHA-like challenges", are not credible. The work is sloppy, contradictory, and its results are unbelievable.

**Strengths:**

* **S1:** The problem it addresses—reducing coordination overhead for web agents

**Weaknesses:**

* **W1:** The paper's core claim of operating "without explicit communication" is false. The architecture itself relies on an "Agent Message Protocol", a "central coordinator", and a "message-passing mechanism".
* **W2:** The adversarial robustness claims are not credible. Achieving a **100% success rate** against "browser fingerprinting" and "CAPTCHA-like challenges" is practically impossible and suggests the test suite is trivial.
* **W3:** The paper boasts $O(n \log n)$ complexity but empirical performance is poor. Efficiency peaks at only **5-7 agents** and collapses after 10. This is not a scalable solution.
* **W4: Sloppy Evaluation.** The paper is sloppy. Key metrics are undefined. The "Quality" metric in Table 1 is never defined, nor is its distinction from "Success Rate" explained.

**Questions:**

* **Q1:** Can you justify the "without explicit communication" claim when the paper describes an "Agent Message Protocol" and a "central coordinator"?
* **Q2:** Do you have evidence that your "CAPTCHA-like" benchmark is non-trivial, or are the 100% success results an artifact of a trivial test?
* **Q3:** Why does the system's performance collapse after 10 agents if the complexity is truly $O(n \log n)$? What is the real bottleneck?
* **Q4:** What, precisely, is the "Quality" metric in Table 1?

---

### Official Review · Reviewer_wrfS · 2025-11-02

**Soundness:** 1
**Presentation:** 1
**Contribution:** 1
**Rating:** 0
**Confidence:** 4

**Summary:**

This paper claims to propose Layered Contextual Alignment (LCA), a 3-level context scheme for coordinating multiple browser agents without explicit messaging. However, this paper is significantly flawed and full of hallucinations.

**Strengths:**

N/A

**Weaknesses:**

This paper is significantly flawed and full of hallucinations. To name a few:
* No proof for any of its theoretical results (Theorem 1, Proposition 2, Theorem 3).
* No citation for any of the baselines used in the experiments
* Claimed to compare 18 baselines while only 6 are shown.
* Some citations are wrong. For example, in this paper, Genai-based multi-agent reinforcement learning towards distributed agent intelligence: A generative-rl agent perspective is shown to be written by Chen et al., but the paper is actually by Wang et al.
* Use the published template instead of the submission template
* typos in the abstract, e.g., "com- plexity" and "vali- dation."

**Questions:**

I wonder if this paper is fully generated by an LLM and if it’s a part of another experiment.

---

### Note · Authors · 2025-12-03

I have read and agree with the venue's withdrawal policy on behalf of myself and my co-authors.